# Functionalization of Graphene Derivatives with Conducting Polymers and Their Applications in Uric Acid Detection

**DOI:** 10.3390/molecules28010135

**Published:** 2022-12-24

**Authors:** Mirela Văduva, Mihaela Baibarac, Oana Cramariuc

**Affiliations:** 1National Institute of Materials Physics, Atomistilor Street, No. 405 A, Ilfov, 077125 Magurele, Romania; 2IT Centre for Science and Technology, Av. Radu Beller Street, No. 25, 011702 Bucharest, Romania

**Keywords:** graphene oxide, reduced graphene oxide, conducting polymers, non-covalent functionalization, charge transfer, uric acid, electrochemical synthesis

## Abstract

In this article, we review recent progress concerning the development of sensorial platforms based on graphene derivatives and conducting polymers (CPs), alternatively deposited or co-deposited on the working electrode (usually a glassy carbon electrode; GCE) using a simple potentiostatic method (often cyclic voltammetry; CV), possibly followed by the deposition of metallic nanoparticles (NPs) on the electrode surface (ES). These materials have been successfully used to detect an extended range of biomolecules of clinical interest, such as uric acid (UA), dopamine (DA), ascorbic acid (AA), adenine, guanine, and others. The most common method is electrochemical synthesis. In the composites, which are often combined with metallic NPs, the interaction between the graphene derivatives—including graphene oxide (GO), reduced graphene oxide (RGO), or graphene quantum dots (GQDs)—and the CPs is usually governed by non-covalent functionalization through π–π interactions, hydrogen bonds, and van der Waals (VW) forces. The functionalization of GO, RGO, or GQDs with CPs has been shown to speed up electron transfer during the oxidation process, thus improving the electrochemical response of the resulting sensor. The oxidation mechanism behind the electrochemical response of the sensor seems to involve a partial charge transfer (CT) from the analytes to graphene derivatives, due to the overlapping of π orbitals.

## 1. Introduction

Uric acid (UA) is an important biomolecule in the human body, due to its connection with the appearance of certain diseases: an increased level of UA is the major cause for the onset of illnesses, such as gout, hyperuricemia, Lesch–Nyhan syndrome [1], diabetes, kidney, and heart disorders, which can irreversibly affect human individuals. UA, as the chemical compound 7,9-dihydro-1 H-purine-2, 6, 8-(3H)-trione, is the main product of purine metabolism. All mammals, except for the primates, are uricolytic organisms, having the capacity for enzymatic oxidation of UA to allantoin, due to the presence of the enzyme urate oxidase. Significant variations in the UA level are due to an increased catabolic rate or as a consequence of a dysfunction in the elimination route, leading either to increased production of UA or an accumulation of it in different parts of the body. Diagnostic confirmation is achieved through monitoring of UA levels in plasma, urine, and saliva samples.

As UA coexists with many other biomolecules inside the human body, such as AA and DA, the main problem regarding its detection arises from the latter’s interference with UA, whose concentration exceeds the concentration level of DA by a hundred times [2].

To overcome these difficulties, many studies have focused on synthesizing new materials which can be employed to modify electrodes, including binary [2] and ternary [2,3] nanocomposites based on carbon structures, CPs, and metallic NPs [4].

Composites based on CPs and carbon NPs (CNPs) [5] decorated with metallic oxides [4] and/or metallic NPs [6] have been successfully used for the selective and sensitive detection of an extended range of biomolecules of clinic interest, such as UA, DA, AA, adenine, guanine, epinephrine, norepinephrine, and so on.

Due to their special properties, nanocomposites can significantly improve the working parameters of sensors, when compared to the individual components [7].

In this work, we focus our attention on composites based on CPs of the Polypyrrole (PPY), poly(3, 4-ethylenedioxythiophene) (PEDOT), polyaniline (PANI), and polyimidazole (PIm) type, as well as CNPs of the graphene oxide (GO), reduced graphene oxide (RGO), and graphene quantum dots (GQDs) type.

Each of the main components within the composite material has a major contribution to sensor performance enhancement; more precisely, regarding the improvement of the electrocatalytic property as well as the sensitivity and selectivity of the electrochemical response. For example, CPs provide the advantage of increased active surface area, in some cases ensuring a porous structure of the electrode, as is the case for PPy [5], which enhances the stability and adhesion properties of the electrode surface. Due to the abovementioned properties, the active specific area facilitates electronic transfer and improves the adsorption of the target bio-analytes at the ES by promoting electro-oxidation of the adsorbed molecules. Monomers that readily polymerize on the ES, assisted by a dopant which helps to increase conductivity, leads to CPs with the enhanced catalytic activity. The class of CPs includes many representatives with very good conductors that help increase conductivity, such as PPy [5,6], PEDOT [8], PIm [9], and others [10,11].

PPy, which has a conjugated structure, is one from the most important CPs and has been successfully used in the electro-detection of various biomolecules [12,13,14]. This is due to several of its properties, including high conductivity, ease of preparation, and good stability. Furthermore, PPy can be electrochemically oxidated to OPPy (over-oxidated PPy), incorporating a high number of carbonyl groups after the oxidation process [15], thus providing unique cationic selectivity, excluding the anionic species to reach the level of ES [12,16] selectivity which is desirable for the detection of biomolecules including DA and UA, among others. Over-oxidation of PPy can lead to a disruption of conjugation and, as such, although OPPy is a good ionic conductor with ion-selective properties, it may also present non-conductive electronic properties [13,16], which could diminish the electrochemical signal collection.

Overall, despite its ease of synthesis, good catalytic activity, low cost, good polymerization yield, and significant chemical, thermal, and mechanical stability, together with good adhesion and high surface area, PPy has also important drawbacks, including high charge transfer resistance and low conductivity [17], which can easily be addressed by combination with conductive carbon structures (i.e., incorporation into hybrid structures).

Another CP often used for electro-detection of UA is PANI. This polymer provides a good matrix for different inorganic semiconductors and carbonic materials, due to its high relative conductivity, chemical and electrochemical stability, and other advantages such as ease of preparation, reduced cost of monomers and economic process of fabrication, reversible redox behavior, and thermal stability. The most active form of PANI has been reported to be emeraldine, due to its high active area and low charge transfer resistance (R_CT_) [17]. However, one important problem remains; namely, sufficient conductivity for applications in sensors field, making it suitable for combination with graphene derivatives or other highly conductive structure which, on their own, cannot allow for a uniform layer deposition on the ES with enhanced selectivity and, further, with extended electro-catalytic activity.

The study conducted by Ghanbari, K. et al., supported by that of Pihel, K. et al. [4,13], has revealed that, by modifying the working ES with composite material based on PANI and graphene, the performance of the sensor could be improved. This fact was translated through an increase in the electronic transfer rate as a consequence of the strong electronic interaction which takes place between graphene and PANI.

PEDOT is another of the most-studied nanostructured polymers, due to its high electric conductivity [18], transparency [19], and high-quality film preparation [20]. As has been reported by many research groups, including that of Reddy, S. et al., PEDOT can be successfully used as an electrochemical sensor in numerous applications, mainly due to its capacity to increase the electrode’s specific area [21,22].

Still, PEDOT alone, with its high conductivity at room temperature, electrochemical reversible behavior, and important thermal and chemical stability, has poor cycling stability and low capacity to form charge transfer channels within its structure, providing low catalytic activity. [23]

Besides these conductive polymers, PIm has been also reported, as pristine or in combination with carbon nanostructures, and tested for UA detection. Polymerized PIm creates a uniform layer on the ES, which is chemically stable and has controllable thickness [24], while the over-oxidation process improves its permselectivity and antifouling process. The combination of PIm with highly conductive compounds can improve its electro-catalytic response [9].

Overall, CPs lead to improvements in the sensitivity and stability of sensors while, at the same time, ensuring the good mechanical strength of the final structure of the composite. The combination of high conductivity and biocompatibility, together with the capacity to immobilize biomolecules on the ES for a long period of time while maintaining their full activity, indicate the high potential of CPs for biosensor construction.

On the other hand, carbon nanostructures present controllable electronic, hydrophilicity (e.g., GO and GQDs), stability, and catalytic properties, due to the functional groups on their surface, as well as high biocompatibility and large surface areas, providing mechanical strength and high conductivity to the composite structure [25].

From the category of graphene derivatives used in the electro-detection of UA, GO is often used, due to its active surface area, water-dispersion capacity and hydrophilicity, thermal and mechanical properties, versatility, and ease of surface modification. A major disadvantage of GO is the tendency of stacking, thus restoring the graphite structure; this disadvantage can be overcome, for instance, with the help of CPs [5,6].

Various functional groups rich in oxygen at the GO surface allow for good dispersion of GO in aqueous solution; however, its conductivity may be lost due to the large discontinuity in the sp^2^ hybrid carbon network. This is another reason why the combination of GO and CPs has high potential, particularly when referring to materials designed for electrode modification in order to achieve better performance in electrochemical detection.

Another graphene derivative with promising applications in electro-detection for a wide range of analytes is RGO, which possesses good properties for sensors, either by itself [26] or as a component of composite materials from the working ES [4,27,28,29]. Due to its high conductivity and significant charge mobility, RGO allows for an increased sensitivity of detection—one of the most fundamental properties of a sensor—by accelerating the CT rate together with a high specific surface area [26]. Another advantage of using RGO is the high number of active sites, as a consequence of the numerous functional groups on its surface, contributing to a uniform deposition of the conductive polymer. Further, the π–π stacking interaction between the CPs and the RGO layer contributes to an enhanced composite conductivity, facilitating charge transport and determining shorter routes for ion diffusion. A significant shortcoming of RGO is its tendency to stack sheets together, leading to a smaller active surface area, which can be overcome by mixing with CPs, which play a stabilizing role in hindering RGO layer aggregation and provide better performance of the working electrode on which the composite material has been deposited.

In the study conducted by Yola, M.L. et al., it has been reported that GQDs, a form of graphene derivatives, when used as part of the modified electrodes used for UA detection, very good response was achieved [30]. GQDs correspond to zero-dimensional (OD) structures with diameter smaller than 10 nm, exhibiting quantum confinement effects and significant edge effects. [31] The main properties of GQDs are (a) high solubility in H_2_O; (b) low toxicity; (c) bio-compatibility; (d) photoluminescence in visible range; and (e) tunable bandgap. All of these properties recommend them for use in various applications, such as bio-imaging [32,33].

Further, composites based on CPs and graphene derivatives such as GO, RGO, and GQDs have been synthesized mainly using three methods: (i) mixing of the constituents, including mechanical mixture [34] or electromagnetic stirring of the mixing solution [35]; and (ii) chemical or (iii) electrochemical polymerization of the monomer in the presence of graphene derivatives [36,37,38,39]. The prevalent method is electrochemical synthesis [40]. Composites can be prepared through the simultaneous [41] or alternative [4] deposition of CPs and graphene derivatives. The major advantage of using this method is the good control of the film thickness, achieved by varying certain parameters such as work potential, current intensity, scan speed, deposition time, and number of cyclic voltammograms. Furthermore, the deposition process is carried out directly on the ES, using small amounts of analytes, with minimal losses.

The electrochemical method can also be used for UA determination, due to important advantages such as rapid detection, low preparation costs, reduced amounts of production materials, and ease of handling, together with the sensitivity and selectivity of the modified electrode [42]. Such electrochemical detection is usually carried out on flat electrodes made from carbon-based materials, such as glassy carbon [5,43], carbon fiber paper [6], carbon paste electrode [44], or ionic liquid-modified screen-printed carbon [45].

In order to increase the sensitivity, the ES is sometimes modified with metallic NPs, carbon nanotubes, carbon monoliths, quantum dots, graphene and its derivatives, and/or CPs [2]. According to the studies reported in the last 6 years, the most promising results for achieving the highest sensitive electrochemical signal on a modified electrode, with linear response in a large range of concentrations and the lowest limit of detection (LOD), turned out to be composites based on carbonic structures and CPs, decorated with metallic NPs [1,3,4].

To highlight the progress recorded over the past ten years in the field of sensorial platforms for UA detection, in the following sections, the main synthesis methods of composites based on CPs and graphene derivatives of the type GO, RGO, and GQDs, as well as aspects of the interactions between components inside the composite, in correlation with characteristics of the sensors modified with these composites, are summarized.

## 2. Chemical and Vibrational Properties of the CPs/GO Composites

### 2.1. Chemical and Vibrational Properties of GO/PIm Composites

The main advantage of using GO/PIm composites is their fast preparation, which can be performed directly on the ES. They can be synthesized through electrochemical co-deposition from a suspension of GO and imidazole (Im). After deposition, the composite can be easily electrochemically over-oxidized. During this process, multiple functional groups rich in oxygen are introduced into the imidazole unit, thus improving the permselectivity and sensor response [9].

A GO/PIm composite has been prepared, according to the electrochemical method described by Liu, X. et al., right on the GCE ES, from 0.3 moles of imidazole dispersed in a mixture of 3 mL GO suspension (5 mg/mL) and 0.3 moles of surfactant (sodium dodecyl sulfate; SDS), with a potential range between −0.2 and 0.8 V. The composite material, PIm–GO, was then oxidized in 0.1 M phosphate-buffered saline (PBS) medium (pH = 3) at +1.8 V for 250 s [9].

When analyzed by FTIR spectroscopy, the PImox–GO composite presented IR bands characteristic of both PIm and GO. Therefore, the IR bands assigned to the C=C vibrational mode of the aromatic ring and the C=N– stretching vibration, located at 1642 and 1468 cm^−1^, respectively, were attributed to PIm, together with the bands situated at 2850 and 2918 cm^−1^ belonging to the stretching vibration of C–H bonds [46,47]. In addition, the corresponding C=C bond band (1642 cm^−1^) in the PIm–GO spectrum was stronger than that for PIm. This, together with the new bands that emerged at 1204 and 3435 cm^−1^, attributed to the C–O and O–H bonds of GO, respectively [48], demonstrated the formation of the nanocomposite.

### 2.2. Chemical and Vibrational Properties of GO/PPy Composites

To improve its electro-conductivity, GO has been combined with PPy, in this way promoting the electrochemical response of the modified sensor due to the π–π stacking between GO layers and PPy rings [6]. The nanocomposites were co-deposited on carbon fiber paper (CFP) using the potentiostatic method, followed by the deposition of AuNPs on the composite surface through CV. The co-deposition was performed using a mix of GO suspension (1 mg/mL), pyrrole monomer (50 mM), Na_2_SO_4_ 0.1 M, and 20 mM SDS, ultrasonicated for 30 min at constant potential of +0.7 vs. Ag/AgCl, using a three-electrode configuration consisting of Ag/AgCl as the reference electrode, Pt wire as the counter electrode, and CFP as the working electrode.

The non-covalent functionalization pattern is also present for the GO composite with PPy (see Figure 1); in this case, the Py monomers are attached onto the GO surface through π–π interactions, hydrogen bonds, and VW forces [49]. After attaching to the GO surface, the pyrrole monomers were electrochemically oxidated, without stacking as a multiple layer structure. The structure of GO deposited on the electrode resembled wrinkled paper with a high active area, while PPy deposited on the top of GO could not be distinguished due to its dispersion on the GO surface, rather than stacking and overlapping as multiple layers. This was considered to occur due to the strong interaction between the GO platform and Py dissociative monomers. In contrast, the pristine PPy deposited on CFP appeared different to the composite, presenting an agglomerated structure.

The characteristic bands of GO and PPy were revealed in IR absorbance spectra, with the bands situated at 1730, 1614, 1217, and 1051 cm^−1^ being assigned to the stretching vibration of C=O, C=C sp^2^ vibration, o the stretching vibration of C–O bond from the epoxy group, and the stretching vibration of C–O from the alkoxy group, respectively, together with a large band situated at 3199 cm^−1^ corresponding to O–H stretching vibration; whereas the IR bands at 1545 and 1470 cm^−1^ corresponded to the stretching vibrations of C–C and C–N from the Py ring, respectively [50]. The bands at 1178 and 921 cm^−1^ corresponded to the doped state of PPy [51], while the sharp band at 1047 cm^−1^ was assigned to the deformation vibration of C–H bonds and stretching vibration of N–H bonds [52]. The bands corresponding to GO (1792 cm^−1^) and PPy (1545 and 1470 cm^−1^) were also visible in GO/PPy and AuNPs@GO/PPy spectra. In the XPS spectrum of AuNPs@GO/PPy, peaks at 288.5, 286.9, 258.2, and 284.6 eV were observed, corresponding to C=O, C−O, C−N, and C−C bonds, respectively. Maxima fitting revealed that the GO structure was rich in functional groups with oxygen content. The XPS N1s spectrum was decomposed in two maxima, located at 401.5 (−N^+^−) and 399.7 eV (−NH−) [53], being considered as further proof of composite generation.

A PPy–GO composite was prepared by in situ chemical polymerization. Py (30 µL) was added to GO (40 mL, 0.5 mg/mL) and an oxidant solution (0.32 g FeCl_3_ × 6 H_2_O). After 4 h, on an ice bath in dark conditions, a black suspension of PPy/GO was obtained. Over 10 mg of PPy/GO powder and 10 mg of polytetraphenylporphyrin (p-TPP) were added into 4 mL of N,N-dimethylformamide (DMF). The resulting mixture was ultrasonicated for 2 h, in order to obtain a stable suspension of p-TPP/PPy/GO composite material. Through deposition of the resulting composite on the GCE ES, the sensor for UA detection was prepared [5]. The PPy/GO nanocomposites presented a structure of micro-pores, thus determining an increase in active surface of the electrode. Pyrrole molecules were adsorbed on the GO surface through π interactions and electrostatic adhesion forces. The adsorption process was facilitated by a high number of functional groups with oxygen (i.e., hydroxy, carboxy, and epoxy groups), as well as active centers, on the GO surface. After polymerization, the PPy layer covered the GO surface uniformly, hindering the agglomeration of multiple layers of GO. Furthermore, the cross-linking created through the introduction of p-TPP indicated a homogeneous mixture of p-TPP and PPy–GO, leading to a uniformly distributed nanocomposite. In the p-TPP/PPy/GO, there was a non-covalent functionalization between GO and PPy, the synergic action of both main components providing high electrocatalytic properties. GO plays a fundamental role in the initiation of the in situ polymerization process of Py monomers, due to its surface being rich in functional groups with oxygen, whereas the PPy film maintains the high surface area of GO by preventing recovery of the graphite structure. According to the FTIR spectra recorded for GO, PPy/GO, and p-TPP/PPy/GO (Figure 2), the signature of GO from the PPy/GO composite was revealed through the IR bands situated at 1738, 1397, and 1092 cm^−1^, attributed to the stretching vibrations of C=O in COOH, and C–O in C–OH and C–O–C, respectively [54,55]. Another band situated at 1635 cm^−1^ was attributed to the stretching vibration of C=C which, together with the bands located at 1401 and 1090 cm^−1^, assigned to C–OH (1397 cm^−1^) and C–O–C (1092 cm^−1^), respectively, complete the GO IR spectrum. The signature of PPy was indicated by the bands situated at 1695, 1041, and 1573 cm^−1^, corresponding to the stretching vibrations of C=N and C–N from the Py ring [56], and to the symmetric stretching vibration of the –CH_2_ bond [54,57]. These results revealed the coverage of GO nanosheets with PPy. The ratio of the D and G Raman band intensities of GO and PPy/GO were similar (i.e., 1.05), indicating the coverage of the GO sheets with PPy had no significant effect on the GO structure, the interaction between the two components being rather weak and, thus, described as a non-covalent functionalization.

The synthesis of the p-TPP/PPy/GO hybrid was verified through Raman spectroscopy. According to Dai, H. et al., the lines associated with the stretching vibration of the ¼ pyrrole ring and phenyl ring (1326 and 1435 cm^−1^, respectively) were emphasized and accompanied by an increase in G band intensity [58] (Figure 3). The PPy-GO composite was characterized by a highly specific area and excellent electric conductivity, while the p-TPP microspheres within its composition can hinder nanocomposite stacking during thermal treatment, thus contributing to the synthesis of a three-dimensional structured nanocomposite.

### 2.3. Chemical and Vibrational Properties of GO/PEDOT Composites

Furthermore, GO can play the role of a dopant in certain composites, balancing the charges; for example, in the case of PEDOT/GO, it contributes to balancing the positive charges from the PEDOT backbone by increasing the amount of functional groups with oxygen [8,59,60].

In order to improve the GO conductivity, which is weakened by the sp^2^ interrupted network of C, new composites with PEDOT were tested, and the synergic effect of both PEDOT and GO led to improved electrocatalytic properties of the resulting sensor [8]. A small amount of GO suspension (1 mg/mL) in deionized water mixed with 3,4-ethylenedioxythiophene (EDOT) monomer (10 mM) was deposited on ITO and covered with Whatman paper, in order to obtain a thin layer electrochemical cell. The piece of paper retains not only the reactants, but also connects the electrode system. The electrochemical polymerization was carried out using chronoamperometry, by applying a potential of 1.2 V for a period of 150 s. At the end, a PEDOT–GO/ITO sensor was obtained [8]. The positive charge of the resulting polymer can attract negatively charged groups from the GO surface and, so, the interaction taking place between the two components is described as non-covalent functionalization [8] (see Figure 4 below). During the polymerization process of the PEDOT–GO nanocomposite on ITO substrate, the composite benefits from the mechanical support of the GO layer, which provides increased stability. Analysis of the composite surface morphology revealed that the PEDOT–GO film was uniformly distributed in a random form, appearing as a porous and rough network. The free space between the layers/sheets is favorable for electron exchange between the biomolecules and the electrode substrate. Overall, combining the components PEDOT and GO resulted in an increase in the sensitivity of the sensor [8].

The changes observed in the UV–Vis absorption spectra of the composite, regarding the shift of the band assigned to PEDOT oligomers from 344 nm [61] to 356 nm, together with the formation of a new broad band located between 600 and 800 nm, were assigned to polarons and bipolarons originating from the partially doped PEDOT [62]. Sensors based on PEDOT and GO have also been prepared by a different method—namely, co-deposition—from a mixture of EDOT 10 mM solution and GO suspension (1 mg/mL), in the potential range between −0.2 and −1.2 V, during 20 CV, and at a scanning rate of 100 mV/s.

Different electrolytes, such as PBS, GO/PBS, and GO, can be used to obtain optimal combinations of the main components which could activate the electro-catalytic property of the electrode. Thus, the preparation of PEDOT–GO/GCE, according to Li, D et al. [59], was performed in two ways: first, the polymerization process was initiated by an oxidant; second, the initiation was conducted through electro-polymerization and the resulting polymeric film could be oxidated through further electro-polymerization [20]. Among different electrolytes for EDOT electro-polymerization on the GCE surface, PBS, GO, and GO in PBS were considered, one at a time, and it was seen that, in the presence of GO as a unique electrolyte, the current amplitude increased with the number of cycles and the initial oxidation potentials were shifted to negative values, suggesting the growth of the PEDOT–GO layer. At the end of the electro-polymerization process, the formation of a metallic film with blue shine confirmed the synthesis of the PEDOT–GO hybrid film. According to Li, D. et al., GO plays a crucial role in promoting nanocomposite formation, providing both electrolyte support for the ion conductor and counterion in the polymer doping process when the positive charges of the polymer have been neutralized. The great active surface of GO provides numerous bonding sites for deposition of the PEDOT layer. On the other hand, PEDOT improved the weak conductivity of GO, and the synergic effect of both components of the composites could lead to a significant improvement in the electrochemical and catalytic properties of the PEDOT–GO composite.

### 2.4. Chemical and Vibrational Properties of GO/PANI Composites

The last category of CP–GO composites is represented by PANI–GO nanocomposites. The materials have been synthesized using an in situ chemical polymerization method. Polymerization was initiated by adding an oxidant (ammonium peroxydisulfate, 0.25 M, in 50 mL of 1 M HCl aqueous solution), in a dropwise manner, to 0.2 M aniline (ANI) dissolved in 100 mL HCl solution (1M), under continuous stirring in acid medium until the characteristic green color of polyaniline (PANI) emeraldine salt appeared. For the preparation of the PANI–GO composite, ANI was added to a GO dispersion in an acidic medium until it was completely dissolved. Then, under stirring, a solution of ammonium peroxydisulfate in 1 M HCl was added, in a dropwise manner. After a couple of minutes, the mixture changed color to green, serving as an indicator for the polymerization reaction.

In the end, the composite was filtered and washed with water, ethanol, and acetone until it became colorless, and was then dried in vacuum at 60 °C. The synthesized PANI–GO composite was drop-casted on the ES and dried for 5 h at room temperature. Comparing all FTIR spectra recorded on GO, PANI, and PANI–GO composite (Figure 5), the main characteristic bands corresponding to GO and PANI could be seen, as follows: the stretching vibration of C–O–C (1050 cm^−1^), the stretching vibration of C–OH (1242 cm^−1^), the deformation of O–H (1384 cm^−1^) from the C–OH group, the stretching vibrations of C=C (1630 cm^−1^), C=O (1732 cm^−1^) from the –COOH group, together with a broad and intense peak assigned to stretching vibration of O–H located at 3409 cm^−1^, indicating the existence of a large number of adsorbed water molecules. Regarding the signature of ANI in the FTIR spectra, it can be recognized according to the stretching vibrations of quinonoid ring (C=N) located at 1567 cm^−1^, of benzenoid rings (C–C) located at 1483 cm^−1^, of C–N stretching situated at 1387 cm^−1^, together with a C–N vibrational stretching of a secondary aromatic amine at 1294 cm^−1^, and large bands assigned to the in-plane and out-of-plane bending of aromatic C–H located at 1126 and 811 cm^−1^, respectively [63]. By comparing the three IR spectra recorded for each component and the composite, some differences appeared as distinctive elements in the spectra of the composites. For example, bands situated at 1656 cm^−1^, assigned to the amide group of the PANI–GO composite, appeared during the interaction between the carboxylic group from the edge of the GO nanosheet with the amino group of ANI, while the band at 2926 cm^−1^, assigned to CH_2_ groups, indicated the restoration of the carbon basal planes as a consequence of the reduction process [64]. All of these results, taken together with the red shift recorded for the IR band assigned to the C=O vibrational mode from 1732 to 1743 cm^−1^, suggest the partial reaction of the carboxyl groups from the edges of the GO surface with the amino group of the ANI.

Of the multiple roles it plays, GO also acts as a dopant in the PANI–GO composite, as demonstrated by the widening of the D band, with respect to the GO sheet Raman spectrum after polymerization took place, due to the in-plane bending vibration of C–H molecules. As can be seen from the Raman spectra recorded for GO, PANI, and the PANI–GO composite, there is a shift towards higher wavenumbers, reported for D band corresponding to the GO spectrum and a shift of the Raman lines from the PANI spectrum from 1347 and 1585 cm^−1^ to 1366 and 1590 cm^−1^ (Figure 6). All of these changes suggest that GO was successfully introduced into PANI, and there exists an interaction between the components of the composite [65]. It is likely that an electrostatic interaction takes place between the carboxylic group of GO and the positive charges from nitrogen atoms belonging to the PANI backbone, together with π–π interactions between the aromatic rings and hydrogen bonds between the OH functional groups from the GO surface and the hydrogen from the N^+^ of PANI.

The type of interaction between the components of the composite also determines the morphology, which changes radically: passing from agglomerated fibers mixed with granular particles reported for PANI alone, to smooth surface fibers with layered structure attached to the GO surface in the case of the composite [63]. Considering the information obtained from characterization performed according to FTIR spectroscopy, Raman scattering, and SEM, the main conclusion regarding the interaction between GO and PANI in the composite structure is that there is a combination of electrostatic interactions, hydrogen bonding, and π–π interactions [65] (see Figure 7).

### 2.5. Overview of Sensorial Platforms Based on the CP/GO Composites

In various composites, GO plays multiple roles, including as an ion conductor or counterion dopant, captured inside the positively charged polymeric film during electro-polymerization, contributing to an increased number of functional groups with oxygen, as well as behaving as a support material (e.g., providing a large number of active nucleation sites for PANI growth). GO provides mechanical support and stability to the composite structure and hinders the polymer from forming multiple layers for stacking—thus providing uniform deposition of the polymeric film—due to its nucleation centers. The polymers also hinder the GO layer from stacking together and restoring the graphitic structure. Furthermore, due to their specificity and the fact that they contain a variety of functional groups, the polymers provide a large interaction area and contributes to the conductivity enhancement, especially together with metallic NPs.

The most important property of GO is that this substrate possesses multiple functional groups with oxygen [66], such as epoxy and hydroxyl groups (which, according to Lerf and Klinovski [67], are the most predominant on the GO surface). According to experimental results, the chemical groups mentioned above (e.g., epoxy and hydroxyl) can form clusters with functional groups on the edges, such as hydroxyl, carbonyl, and carboxyl groups, coexisting with non-oxidized islands areas of the GO. The density functional theory (DFT) calculations and analysis regarding both the GO structures with cluster functional groups and GO with isolated functional groups have revealed the highly stable structure of GO [68]. Furthermore, the study conducted by Prasert, K. et al. [68] has reported the results of DFT calculations carried out on the basis of the interactions that occur between the GO substrate and isolated biomolecules, such as AA, DA, and UA. Supercells of graphene (with area of 5 × 5 cm^2^) were considered for this study, grafting 5 hydroxyl and 5 epoxy groups onto each of them, using the configurations proposed by Domancich et al. [69]. The functionalization degree was selected such that the GO structures should be energetically stable, the latter being evaluated through energy formation by functional group. The functionalization of graphene with these types of groups—that is, hydroxyl (–OH) and epoxy—involves an exothermic process, where the highest energy consumption recorded was for epoxy group formation [67]. The strongly electronegative oxygen atoms from the substrate were assumed to have the tendency to receive electrons from the π orbitals of the analytes, such that a partial CT takes place from the analytes with high aromaticity to the GO, due to a higher probability for the π orbitals of DA and UA to overlap [67]. The synergic effect of the CP and GO led to a significant improvement in UA detection, as in the case of GO composites with PPy, PEDOT, and PImox [5,6,8,9].

In the case of the PPy–GO composite, cyclic voltammograms were recorded in PBS 0.1 M solution (pH 7.0) containing 1 mM AA, 1 mM DA, and 1 mM UA. Due to the different structures, the analytes interacted differently with PPy, as reflected in the difference between the mass transfer and the electrochemical activity reported for PPy/CFP. On both of the modified electrodes, GO/PPy/CFP and AuNPs@GO/PPy/CFP, the anodic peaks of the investigated analyte were completely separated, accompanied by an increase in the peak current. As a result of the synergic action of the sensor components, distinctly separated peak potential and high peak current were recorded for each analyte in simultaneous determination [9].

The synergic effect of CP and GO has also been supported by another study, published by Dai, H et al., where an electrode modified with p-TPP/PPy/GO was tested for UA determination in phosphate-buffered solution (pH = 7) containing different concentrations of UA in the presence of constant concentration values of AA and DA. The sensor obtained by drop-cast deposition of p-TPP/PPy/GO film on GCE ES presented a linear response in the concentration range (CR) of 5–200 µM and a LOD of 1.15 µM for UA [5].

Similar results have been reported in the study of Tan, C. et al., where three distinct potential peaks corresponding to AA, DA, and UA were recorded on a GO/PPy modified electrode, tested in real urine samples where successive amounts of AA, DA, and UA were added [6]. In this context, considering the reported results, the electrocatalytic activity of the electrodes modified with PPy/GO could be described as dependent on multiple factors, such as the selective interface provided by the hydrogen bonds between GO and PPy with the functional groups of UA, leading to an increased concentration of analytes on the ES and, therefore, lower over-oxidation potential; the π–π interactions and the hydrogen bonds between GO and PPy, which speeds up electron transfer during the oxidation process, thus enhancing the current intensity; and last, but not least, increased conductivity due to AuNPs, facilitating the oxidation of analytes. The most relevant result supporting these affirmations was represented by a low Rct value (of just 5.18 Ω) for the GO/PPy/CFP electrode, in contrast with that for simple CFP (540.9 Ω), which was further improved with the addition of AuNPs (3.92 Ω), revealing the good electronic transfer performance of the modified electrode [6].

Further, Huang, X. et al. reported similar results for a PEDOT–GO/ITO sensor tested in artificial spit samples at 7.5 pH, with a linear response within the CR of 2–1000 µM and a 0.75 µM LOD [8]. The peak potential was recorded at 0.27 V for the oxidation process of UA controlled by diffusion [8].

Another study based on the PEDOT–GO/GCE sensor activity has revealed the importance of the active specific surface on the electrocatalytic activity of the sensor [59]. An electrode modified with a PEDOT–GO composite sensor presented a specific active surface, wider than that reported for PEDOT/GCE (0.1385 cm^2^ vs. 0.0787 cm^2^) and a CT resistance of 0.2 Ω, compared with 77 Ω for PEDOT/GCE (see Table 1).

Under the optimized DPV experimental conditions, the peak current of UA showed a linear response over the range of 2–18 mM (R^2^ = 0.9902), with an LOD of 0.2 mM [59].

The interaction between GO and CPs, described through π–π interaction, hydrogen bonds, and VW forces, is governed by non-covalent functionalization. The functionalization of GO with PPy has been shown to speed up electron transfer during the oxidation process, thus improving the electrochemical response of the sensor. The oxidation mechanism, which underlies the electrochemical response of the sensor, seems to be governed by a partial CT from the analytes to GO due to the overlapping of π orbitals.

The contribution of all components of the composite leads to a change in the morphology of the composite film (Figure 8). For example, the surface of the GO/PImox composite became rougher and more available to retain the analytes, while the PANI–GO surface was either smoother and more sensitive when in contact with the analytes, or maintained the look of wrinkled paper, covering the GO sheet uniformly and providing high selectivity of the sensor. A uniform distribution and compact deposition of PEDOT NPs together with GO sheets have also been reported, resulting in a structure with the semblance of wrinkled paper when a GO–EDOT suspension was used for the hybrid film synthesis [59].

All types of composites based on GO and PCs revealed uniform deposition of the polymeric film on GO layer surface, highlighting the wrinkled paper appearance in the case of PEDOT–GO and PPy–GO, and a coarser structure for PImox–GO with the morphology of agglomerated fibers mixed with granular particles, especially in the case of PANI–GO. These results provide good evidence for the polymers improving the active surface area and facilitating a more efficient adsorption and oxidation process of the target biomolecules, as detailed in Table 2.

## 3. Chemical and Vibrational Properties of CPs/RGO

The methods of synthesis used to obtain this type of composite range from the simplest and fastest, such as electrochemical methods [4,70], to complex routes for preparation of composites; for example, through a hydrothermal route [27]. Among the CPs, the most-reported in combination with RGO to obtain composites used for UA detection are PPy [71], PANI [4,27], and PEDOT [72].

### 3.1. Synthesis and Interaction between RGO and PPy Inside the Composite

According to the study published by Chen, X. et al. [71], composites based on PPy and RGO were prepared starting with the deposition of RGO film on top of the GCE. Next, using a Py solution (0.2 M) in PBS medium (10 mM, pH = 7.4), within the potential range of −0.2 to 0.8 V, the PPy film was deposited on the RGO/GCE surface. The final stage involved PPy film oxidation, performed through an electrochemical process within the potential range of 0–1 V, during 4 cyclic voltammograms in NaOH 0.1 M solution. The electrochemical properties of GCE modified with OPPy (over-oxidated PPy) and RGO were tested using redox samples, including negatively charged Fe (CN)_6_^3−^ and positively charged Ru (NH_3_)_6_^3+^, comprising electroactive compounds of similar size. On PPy/ERGO/GCE, the charge currents increased and the redox maxima disappeared. On the other hand, when using OPPy, within OPPy/ERGO/GCE, the charge currents decreased, the ruthenium redox maxima disappeared at an oxidation potential of 0.253 V, and a reduction potential of −0.237 V was observed. The oxidation maximum increased by 4.7 times and the reduction potential by 9.3 times, compared to GCE alone. According to [71], these results indicated that the ruthenium sample reached the ES and CT took place. The hybrid nanocomposite OPPy/ERGO has ion-selective properties, as the negative charges of OPPy allow only cations to reach the ES and be involved in the CT process. Comparing the electrochemical behavior with both the positively and negatively charged systems, the influence of the PPy over-oxidation process on the electrochemical activity of PPy/ERGO hybrid composite was highlighted. The negatively charged OPPy film led to electrostatic adsorption. On the PPy/RGO film surface, with the help of SEM images, it was observed that small PPy spheres covered the entire surface. This indicated the formation of a 3D polymeric structure through in situ polymerization had occurred on the ES, with the help of a graphene template [73]. After performing the over-oxidation treatment on the ES, a rough, compact, and uniform film was obtained, resembling sheets of wrinkled paper with many edges.

Another method for synthesis of an RGO/PPy composite involved co-synthesis from a mixture of GO (40 mg) and cetrimonium bromide (91 mg), phosphoric acid (15 mL), and Py monomer, which was gradually added to the initial mixture and stirred for about 2 h at 15 °C [29]. Polymerization was initiated by adding ammonium persulfate solution (APS). After 4 h of reaction, the black precipitate was filtered, washed, and dried. RGO/Pd@PPy nanocomposite was obtained by dispersion of RGO/PPy powder in ethylene glycol and 7 mg/mL PdCl_2_ solution. Then, the mixture was exposed to microwaves for thermal treatment for 2 min, following which it was centrifuged and dried at 100 °C under vacuum. The RGO/Pd@PPy NPs was then drop-casted on the GCE surface. According to the Raman spectroscopy analysis, the I_D_/I_G_ ratios reported for RGO/PPy and RGO/Pd@PPy were equal to 0.87 and 0.96, respectively, which was interpreted as proof of the functionalization of RGO with PPy–Pd [29]. Comparing the SEM images recorded on PPy/ERGO before and after the over-oxidation process, it was found that the PPy/ERGO film had a laminated structure before oxidation, with multiple spheres of PPy covering the PPy/ERGO/GCE surface, suggesting the generation of a 3D polymeric structure on the ES. According to the results obtained by Demirkan, B., the ERGO layer, deposited before starting the polymerization of Py, plays the role of a template, guiding the entire process of the PPy film growth. At the end of the deposition process, after the over-oxidation stage, a compact, uniform, and thin hybrid film of PPy/ERGO was formed [29].

### 3.2. Chemical and Vibrational Properties of RGO/PANI Composites

A PANI/RGO composite has been prepared through chemical synthesis from GO, hydrothermally treated with a mixture containing Fe(NO_3_)_3_, SnSO_4_, and PANI, obtained through oxidative chemical polymerization from 500 mg ANI, 5 mg surfactant (Triton X-100), 20 mL ultrapure water, and 10 mL HCl solution (1 M) [27]. According to the synthesis reported by Minta, D. et al., a mixture of ANI and surfactants, in an acidic medium, was ultrasonicated for 30 min at a temperature between 0–5 degrees, after which polymerization was initiated through the addition of ammonium persulfate. The final composite was synthesized by adding 50 mg of GO decorated with tin and iron oxides in ultrapure water and ultrasonicated during 30 min. Then, PANI powder was added, and the reaction took place in an autoclave at 180 °C, for 12 h, under stirring. After several wash/dry cycles, the resulting composite had a mass ratio 50:17:33 PANI:Fe_2_O_3_–SnO_2_:RGO (PFSG).

The PANI/RGO formed through in situ electrochemical deposition presented a mixed morphology, having both the RGO layer structure and the PANI nanofibers distributed on the graphene layer surface, obtained after performing the ANI polymerization in HCl [4]. The main IR bands from the PANI spectrum, located at 1390, 3445, and 2830 cm^−1^, were assigned to the stretching vibrational mode of C–N, N–H, and C–H bonds, respectively. To these, the stretching vibrational mode of quinoid ring, located at 1138 cm^−1^ (N–Q–N–Q) was added. The prominent absorption bands were assigned to C=C deformation from the quinoid ring (1585, 1498 cm^−1^) and to the stretching vibrational mode of the C–N bond of the secondary aromatic amine [63]. In the FTIR spectra of RGO/PANI, the absorption bands from 1560 and 1487 cm^−1^ were assigned to the stretching vibrational mode of C=C bonds within quinoid and benzene structures, respectively [74]. The bands from 1296 and 1242 cm^−1^ were attributed to the stretching vibrational mode for C–N and C=N bonds, respectively [75]. 

These bonds suggest the coverage of RGO sheets with PANI, and the results provide evidence of the functionalization type which takes place between PANI and RGO; namely, covalent functionalization [76] (see Figure 9).

The presence of these bands suggest the coverage of ZnO with PANI/RGO, resulting in a ZnO/PANI/RGO composite, which leads to a shift in the position of peaks and changes in their relative intensity. Regarding the ZnO/PANI/RGO FTIR spectrum, the peaks located at 1560, 1487, and 1390 cm^−1^ shifted towards the higher wavenumbers of 1566, 1488.9, and 1394 cm^−1^, respectively, indicating that the nitrogen atoms from amine groups (NH) and imine groups (N) had bonded with Zn^2+^ through protonation and complexation reactions [77]. The band of high intensity from 1394 cm^−1^ and the wide band from 437 cm^−1^ correspond to the stretching vibration of O–Zn–O [78]. The strong stretching vibrations of C–N^+^ bonds, represented by the Raman line at 1330 cm^−1^, correspond to radical cationic structures, such as the protonated semi-quinoid structure. The lines situated at 1225 cm^−1^ and 1170 cm^−1^ could be assigned to the stretching vibration of C–N bonds in the polaronic units and in-plane bending vibrations of C–H bonds corresponding to bipolaronic shapes, respectively. The Raman lines from 829, 770, 507, and 411 cm^−1^, correspond to the bending vibration of the C–N–C bond, to the quinoid ring deformation, and to in-plane and out-of-plane amine deformation, respectively. The Raman lines from 1485 and 1583 cm^−1^ were assigned to the RGO signature shift from the initial position, meaning that a bond between C–N^+^ and carboxylate group from the RGO sheets was created, which was formed on the basis of the π–π* interactions occurring between PANI and RGO [79]. Confirmation of the incorporation of RGO into the PANI matrix during electro-polymerization was recorded through the two bands at 2770 and 3070 cm^−1^, which were assigned to the respective contributions of the D and G bands, together with the contribution of the 2 D band from the Raman spectra of PANI/RGO composite [80,81].

### 3.3. Chemical and Vibrational Properties of RGO/PEDOT Composites

In the case of composites containing PEDOT and RGO, both constituents (i.e., RGO and EDOT) were introduced into an electrolyte solution and concomitantly deposited on the ES by CV, within the potential range of 1.2 to −1.5 V vs. Ag/AgCl [82]. The pristine PEDOT film presented a granular structure, revealed through SEM analysis; while, after polymerization with the mixture of the GO and EDOT solution, a rough surface was recorded for the PEDOT/rGO film.

According to the FTIR spectra recorded for PEDOT and GO before and after the reduction process, and that of PEDOT–RGO (PrGO), the IR bands characteristic of GO, rGO, and PEDOT were present. More specifically, peaks located at 792, 1318, 1658, and 2796 cm^−1^ were assigned to C–S stretching mode and the C–O–C, C=O, and sp^3^ C–H stretching modes of PEDOT backbone, respectively. On the other hand, IR bands located at 1317 and 3310 cm^−1^ were found to correspond to C=O and O–H groups from the GO structure; which, after the reduction process of GO, decreased in intensity, as the number of oxygenated functional groups was decreased. By analyzing the IR spectrum of the final composite, the presence of peaks from both components suggested the good incorporation of RGO into the PEDOT polymeric matrix.

The most significant changes were noticed in the shifts to higher wavenumbers of the bands assigned to both RGO and PEDOT; for example, the bands located at 3011 and 971 cm^−1^ corresponding to RGO spectrum and the band from 2796 cm^−1^ corresponding to PEDOT, which was shifted to 2844 cm^−1^, strengthened the evidence for the successful synthesis of the PEDOT–RGO composite [16] (Figure 10).

These results suggest coverage of the GO layer with a polymeric film during electro-polymerization, where the π–π interactions and VW forces between the aromatic ring of RGO and PEDOT played a significant role in PrGO film formation. This reveals the important role played by the metallic oxide, which contributes to the increase in the electro-oxidative activity of the sensor. After decorating it with MnO_2_, small grains uniformly dispersed on the PrGO composite surface were observed, which were responsible for enlarging the active surface area, thus emphasizing the wrinkled paper appearance of the RGO layer. A small difference in ratio between the D and G Raman lines was reported for the PrGO/MnO_2_ composite (I_D_/I_G_ = 1.06), compared to the PrGO composite (I_D_/I_G_ = 1), which was caused by the defects introduced into the PrGO structure by MnO_2_.

### 3.4. Overview concerning Sensorial Platforms Based on the CP/RGO Composites

The main CP/RGO composites reported for UA detection are PPy–RGO, PANI–RGO, and PEDOT–RGO. Each constituent of the composite contributes to the enhancement of the electrocatalytic properties of the sensor.

Due to its high conductivity and significant charge mobility, RGO provides an increased sensitivity of detection by enhancing the CT rate, while providing a high specific surface and a high number of active sites due to the numerous functional groups on its surface, leading to uniform deposition of the CP. The CPs also influence the morphology of the composite film (see Figure 11), additionally playing a stabilizing role to hinder RGO layer aggregation and providing better performance of the sensor. Some CPs maintain the wrinkled paper morphology of the RGO template, on top of which they are uniformly deposited or, in some cases, co-deposited through electro-polymerization from GO and monomer dispersion (e.g., PANI, PEDOT), thus providing many adsorption centers.

Besides their high relative conductivity, CPs also provide chemical and electrochemical stability and other additional advantages, such as ease of preparation, reduced cost of monomers and economic process of fabrication, reversible redox behavior, and thermal stability. The π–π stacking interaction between the CPs and the RGO layer contribute to an enhanced composite conductivity, facilitating charge transport and determining shorter ion diffusion routes. The electronic transfer rate may be further enhanced after decorating the composite material with metallic NPs. An example which revealed improved sensor response after introducing a conducting polymer onto its surface has been reported by Minta et al., who regarded ternary composites based on PANI and RGO and highlighted the role played by the CP in increasing the upper limit for the CR (from 150 to 300 µM) and the peak-to-peak separation [27].

Another factor which strongly influences the form of UA—and, therefore, the interaction with composites with the electrode surface—is the pH value of the reaction medium. For example, at a pH value of 6.6, UA has an anionic form, which allows for the immobilization of opposite charges of imines from UA, with the help of the positively charged groups in PANI [83,84,85]. The oxidation mechanism of UA which occurs on the composite surface is due to the interactions which occur between UA and the composite, thus depending on the chemical structure of the analyte. The adsorption of UA involves the interaction between the cyclic ring of UA and functional groups with oxygen atoms (i.e., COO–) from the composite.

When UA oxidation occurs, Minta, D. et al. reported that the potential of the anodic peak shifts to negative values; namely, from 500 to 470 mV. The sensor selectivity was tested taking into account the concentration ratio of DA and UA, which is 10 times higher than that of DA (20–150 µM). In these conditions, the anodic peaks of both analytes were still very well-separated, and they could be clearly distinguished with a peak-to-peak separation of 160 mV, confirming the good selectivity of the sensor for UA and DA detection, with an LOD of 6.4 µM for UA in concomitant detection. For both DA and UA detection, the oxidation process of biomolecules takes place according to the contribution of the protons [83,86].

Furthermore, when combining CP with RGO, a synergic effect arises, facilitating CT at the electrode/analytes interface, thus promoting redox reactions. The abovementioned results have been recorded for composites based on PANI and RGO, with linear response reported between 0.1 and 100 µM, CR of 100–1000 µM, and an LOD of 0.042 µM for individual detection of UA, as well as 0.12 µM LOD in the presence of AA (1 mM) [4]. Another composite based on CP and RGO—namely, PEDOT–RGO—has been mentioned in terms of its good performance in the range of concentration varying between 1–300 µM, and an LOD of 0.19 µM [16]. An additional study reported by the same research group—namely, Tukimin, N. et al.—revealed a good performance and even better LOD when decorating the PEDOT–GO composite with metallic oxide NPs [72]. The PEDOT–GO composite decorated with MnO_2_ was tested for the concomitant detection of UA, DA, and AA, showing high peak current response and a well-separated oxidation potential of the investigated analytes. The best electrochemical response was reported for the PEDOT–RGO composite at pH = 6, within the range 0.3–80 µM and with an LOD of 0.05 µM [72], after interaction between the cyclic ring of UA—namely, the hydrogen bonds established between the OH groups and the nitrogen atoms of UA [87,88]—and the chemical groups containing oxygen (e.g., COO–) in the PEDOT–GO/MnO_2_ (PrGO/MnO_2_) composite.

The properties of flat metallic electrodes used as electrochemical sensors, such as screen-printed electrodes (SPEs), have also been reported in the literature [27], which demonstrated their good electrocatalytic activity regarding UA oxidation.

Summarizing the information presented in this section (see Table 3), it appears that the most promising combinations of CPs and RGO used for UA detection are as follows: (a) PANI/RGO, for which the widest LR has been reported (1–1000 µM) [4]; and (b) PEDOT/RGO, for which the lowest LOD has been reported for UA detection (0.05 µM) [72]. 

## 4. Synthesis and Characteriz4. Chem4. Chemical and Vibrational Properties of CP/Graphene Quantum Dot (GQD) Composites

Various strategies have been used to prepare GQDs, with the most important of them involving (a) the acidic oxidation of various carbon-based materials of GO [89], multi-walled carbon nanotube [90], graphene [91], and graphite [92] type, yielding GQDs with size around of 3.0 ± 1.0 nm, <5 nm, 4 ± 0.2 nm, and 22.4 nm, respectively; (b) electrochemical electrolysis using a graphite rod as the working electrode, with GQDs having size between 2–25 nm reported [93,94,95,96]; (c) ultrasound-assisted methods, which use GO [97], graphite powder [98], carbon fibers [99], few-layer graphene sheets [100], and acetylene black [101] as carbon sources, yielding GQDs varying in size from 6.5 ± 1.2 nm to 3.25 nm, 1–8 nm, and 2–9 nm, respectively; (d) synthesis assisted by beam irradiation of GO [102] or salicylic acid [103], obtaining GQDs with size of 2–10 nm and 3 nm, respectively; (e) chemical exfoliation of graphite [104], MWCNT [105], and carbon fibers [106], resulting in GQDs with size of 14 ± 4.1 – 34 ± 6.9 nm, 3 nm, and 5 ± 2 nm, respectively; (f) solvo-thermal and hydrothermal synthesis with the assistance of corn powder [107] and GO [108], yielding GQDs with size of 20–30 nm and 4–5 nm, respectively; and (g) pyrolysis methods, using citric acid [109] or glucose and urea [110] as a carbon source, resulting in GQDs with a size of 10–20 nm and 3 nm, respectively.

By analogy with GO or RGO, GQDs show two bands in the Raman spectrum, labelled as the D and G bands, assigned to the defects which involve the C atoms with sp^3^ hybridization and E_2g_ vibration mode of the C atoms with sp^2^ hybridization, respectively [111,112,113]. The ratio between the intensities of the D and G Raman bands is around 0.8 [114]. The functional groups of GQDs have been highlighted in FTIR spectroscopy studies. When GQDs were observed, in terms of IR spectra, the bands peaked at 1107 and 1763 cm^−1^, indicating the vibrational modes of the bonds C–O–C and C=O or –COOH– [115].

For concomitant detection of UA, DA, AA and tryptophan (TR), GQDs functionalized with rod bi-metallic NPs of Au–Pt have been synthesized by M.L. Yola et al. [30]. In this order, the authors used 1 mg mL^−1^ of bi-metallic NPs mixed with 0.1 mg mL^−1^ of GQDs in a volumetric ratio of 1:1. GQDs prepared by the acidic oxidation of the graphene layers interacted with H_2_SO_4_ and HNO_3_ for a duration of 15 h, followed by interaction with N-(3-dimethylaminopropyl)-N-ethyl carbodiimide hydrochloride for 10 h, and then with 2-aminoethanethiol for 1 h [30]. The electrode for the detection of UA, DA, AA, and TR was obtained by deposition of the functionalized GQDs with bi-metallic NPs through the drop-casting method onto the GCE surface.

A recent report in 2019 has described a new sensor type of GQDs doped with nitrogen, PANI, and SnO_2_ for the detection of DA in the presence of UA and AA [116].

An example of GQDs functionalized with PANI is presented in Figure 12. GQDs doped with nitrogen (N-GQDs) were synthesized by a hydrothermal method, using citric acid and urea as the reaction mixture, at a temperature of 160 °C. Successively, ethanol was added to the mixture, interacting with H_2_O, followed by dialysis for a duration of 24 h. SnO_2_ NPs were synthesized by hydrothermal reaction using a mixture of K_2_SnO_3_ × 3H_2_O and ethylene glycol, the temperature of hydrothermal reaction being 200 °C for one day. After filtration and water rinsing, it was dried for one day at the temperature of 60 °C. Composite SnO_2_/PANI was obtained by chemical polymerization of ANI in HCl solution containing SnO_2_ and (NH_4_)_2_S_2_O_8_, with the polymerization reaction of ANI taking 3 h at 0 °C. To obtain the SnO_2_/PANI/N-GQDs composite, GQDs were added to the above reaction mixture and the precipitate was filtered, washed with CH_3_OH, and dried at the temperature of 60 °C under vacuum, over the period of a day. A new type of sensor, consisting of GCE modified with GQDs-doped PEDOT for the concomitant detection of AA, DA, and UA, has recently been reported by H.C. Lim [117]. Composite PEDOT/GQD was prepared during the electro-polymerization of monomer—that is, EDOT mixed with LiClO_4_ and GQDs in the semi-aqueous solution of H_2_O:CH_3_CN (1:9 volumetric ratio)—by CV, with the potential range from -0.5 to +1.7 V and potential scan rate of 100 mV s^−1^ [117]. According to the study conducted by Lim, H.C. et al., the Raman spectra recorded on PEDOT–GQDs composites revealed lines at 1351, 1434, 1496, and 1558 cm^−1^, assigned to C_β_–C_β_ stretching, C_α_=C_β_ symmetric stretching [118], and asymmetric stretching of C_α_=C_β_ [118], respectively. A relevant change was noticed in the shift of the lines from 1434 cm^−1^ assigned to the symmetric stretching of C_α_=C_β_ with 12 nm, to lower wavenumbers, in comparison with PEDOT–ClO_4_ (1446 cm^−1^). [117] The red-shift, in this case, was associated with the transition from a double to single C–C bond. This result, together with the enhanced Raman signal of PEDOT–GQD, compared to PEDOT–ClO_4_, suggests a high packing degree between the PEDOT chains. By introducing the GQD anions into the PEDOT backbone as a dopant, the final material—namely, the composite—has an expanded quinoid structure and benefits from extended π-electron delocalization along the PEDOT conjugated backbone [119,120]. Therefore, the morphology of this composite is a porous interconnected 3 D network [117].

UA detection has been carried out using GCE covered by drop casting with an SnO_2_/PANI/N–GQD composite. According to [116], DPV studies indicated that the SnO_2_/PANI/N–GQD/GCE sensorial platform shows three oxidation maxima in the presence of DA, AA, and UA, with potentials at +0.096 V, −0.192 V, and 0.287 V, respectively [116]. The LR of the SnO_2_/PANI/N–GQD/GCE sensorial platform, in the case of DA mixed with AA and UA, is between 5 × 10^−7^ M and 2 × 10^−4^ M [116].

In the experiment conducted by Lim, H.C. et al., GCE modified with PEDOT/GQDs detected AA, DA, and UA in the presence of PBS at a pH of 7.4. Three anodic peaks were observed for the three analytes UA, DA, and AA at 0.311 V, 0.180 V, and 0.055 V [117]. The LR of this sensor was reported to be between 30–1000 μM [117]. For the four sensorial platforms, one can observe that the LOD varied between 2.2 × 10^−7^ M and 10^−9^ M (see Table 4). These values were explained by considering the high surface area of GQDs and higher weight of active sites on the GQDs surface. Another explanation concerning GQDs-based sensors was connected to the conductivity of its composites, to the electron transport with high speed induced by the edge effect, and to the quantum confinement of GQDs. A study from 2016, conducted by Yola, M.L. et al., has reported the CV recorded for Au–Pt NPs/GQDs/GCE, revealing the characteristic maxima for UA, DA, AA, and TR having the potential of anodic oxidation at 0.5 V, 0.3 V, 0.1 V, and 1 V. 

Depending on the pH value of the sodium acetate/acetic acid buffer, the authors reported an increase in anodic currents only for DA and UA, accompanied by a slight variation in the particular cases of AA and TR; furthermore, the optimal pH of the buffer was equal to 4 [30].

Another sensorial platform used for the concomitant determination of AA, DA, and UA was that reported by K. Kunpatee et al., which correspond to a GQDs/ionic liquid (IL)-modified screen-printed carbon electrode (SPCE) [45]. The synthesis method of GQDS used by K. Kunpatee et al. was citric acid pyrolysis. The GQDs/IL–SPCE platform was prepared using a polyvinyl chloride substrate, onto which the reference electrode was printed using Ag/AgCl ink, and the counter and working electrodes formed a patterned Ag layer that was modified with a carbon ink and [BMIM]PF6 IL. Deposition of GQDs onto the working ES was performed by authors after an annealing treatment at 55 °C of the platform with three electrodes [45]. Three anodic oxidation peaks were reported for the GQDs/IL-SPCE sensor at −0.02 V, +0.18 V, and +0.36 V for AA, DA, and UA, respectively [45]. DPV studies have evidenced an oxidation reaction at the interface of the GQDs/IL–SPCE sensor and PBS with pH of 4 containing AA, DA, and UA, as controlled by diffusion processes [45]. The LR of concentration in the individual and simultaneous analysis of UA were 0.5–20 and 0.5–10 μM; those for DA were 0.2–15 and 0.2–6; and c) those for AA were 25–400 in both cases [45].

According to above information, the first application of ternary composites of the type SnO_2_/PANI/N–GQD in the sensor field for the electro-detection of DA, AA, and UA was dated to 2019 [116].

Considering all studies which have reported the functionalization processes of GQDs with CPs and inorganic compounds, new sensors are expected to be tested in the near future, both for the detection of UA and other compounds in the pharmaceutical and medicinal industries. To back up this statement, two examples are highlighted: (a) the case of the l PANI/GQD sensor for calycosin detection, a compound having pharmacological effects, for which the LOD has been reported by Cai et al. to be 9.8 × 10^−6^ mol/L [121]; and (b) GCE modified with GQDs and poly(sulfosalicylic acid) has been used for the electro-detection of estradiol and progesterone (steroid hormones), for which the LOD was reported to be 0.23 nmol L^−1^ and 0.31 nmol L^−1^, respectively [109].

## 5. Conclusions and Outlook

The progress reviewed here demonstrates that the interactions between GO and CPs, described through π–π interactions, hydrogen bonds, and VW forces, are governed by non-covalent functionalization. The functionalization of GO with PPy has been shown to speed up the electron transfer during the oxidation process, thus improving the electrochemical response of the sensor. The oxidation mechanism which underlies the electrochemical response of the sensor seems to be governed by a partial CT from the analytes to GO, due to the overlapping of π orbitals. The adsorption of UA involves an interaction between the cyclic group of UA and chemical groups with oxygen (e.g., COO–) from the composite. On the other hand, regarding the CPs/RGO composites, an important role in improving the sensor response is played by the π–π stacking interaction between the CPs and RGO layer, which contributes to increased conductivity facilitating charge transport and determining shorter ion diffusion routes. The functionalization of RGO with CPs is mostly non-covalent, with the interactions between the components including VW forces, hydrogen bonds, and hydrophobic and electrostatic forces, the characterization results serving as obvious evidence of the covalent functionalization only in the case of the PANI–RGO composite, due to π–π* interactions.

Comparing the performance of the GQDs/CPs composites with those reported for CPs/GO and CPs/RGO, in either classic composite or ternary systems, the results obtained for PPy/RGO recorded the best results, with an LOD of 0.047 µM and an extended LR between 1.4–219 µM, as obtained in simultaneous detection, with the tested samples being similar to real ones.

Regarding the results reported for the best combinations of metallic NPs/GQDs and RGO functionalized with CPs, we may conclude that the field of composites used for electrochemical detection of UA (as well as other important biomolecules) is still open to improvement and innovation. No matter the claimed quality of the prepared sensors for UA detection, none of them have claimed to operate in at least a millimolar range, which is necessary to achieve the concentration level of UA found in the healthy human body.

This problem may be addressed by new synthesis approaches involving new geometries of classical carbon compounds, such as the use of a carbon-based aerogel structure to replace the Pt counter electrode in dye-sensitized solar cell (DSSC) devices. These types of structures, with good adhesion on ES, are highly conductive, provide a high surface area and a fast CT rate to fulfill the demand of sensors, and offer a cheaper and easily obtained alternative to CPs/graphene derivative-based composites.

## Figures and Tables

**Figure 1 molecules-28-00135-f001:**
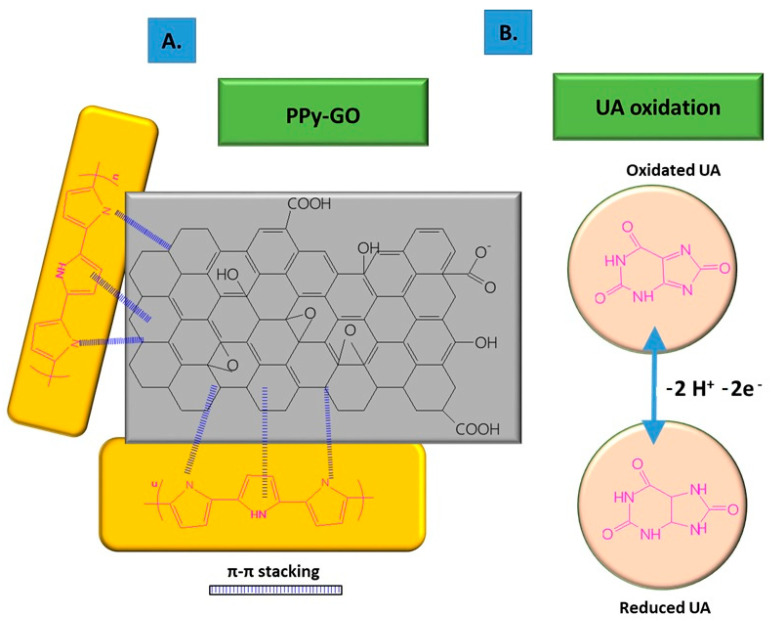
Non-covalent functionalization of GO with PPy inside the composite (**A**); and UA oxidation process at the surface of the composite deposited onto ES (**B**).

**Figure 2 molecules-28-00135-f002:**
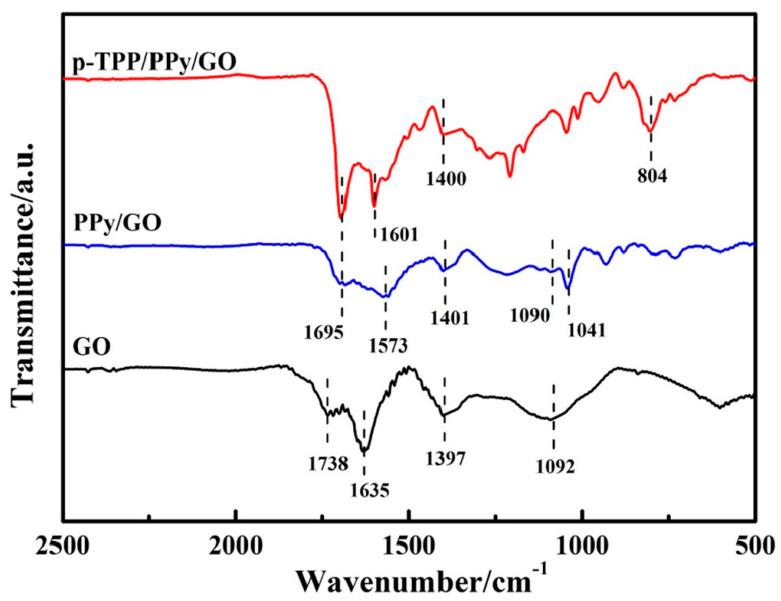
FTIR spectra of the compounds: GO (black curve), PPY/GO (blue curve), and p-TPP/PPY/GO (red curve). Reproduced with permission from ref. [5]; Copyright 2022 Springer.

**Figure 3 molecules-28-00135-f003:**
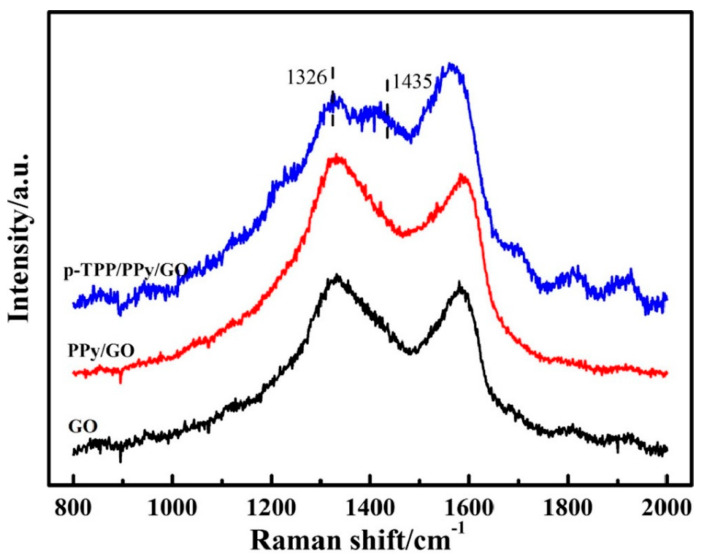
Raman spectra of the compounds: GO (black curve), PPY/GO (red curve), and p-TPP/PPY/GO (blue curve). Reproduced with the permission from ref. [5]; Copyright 2022 Springer.

**Figure 4 molecules-28-00135-f004:**
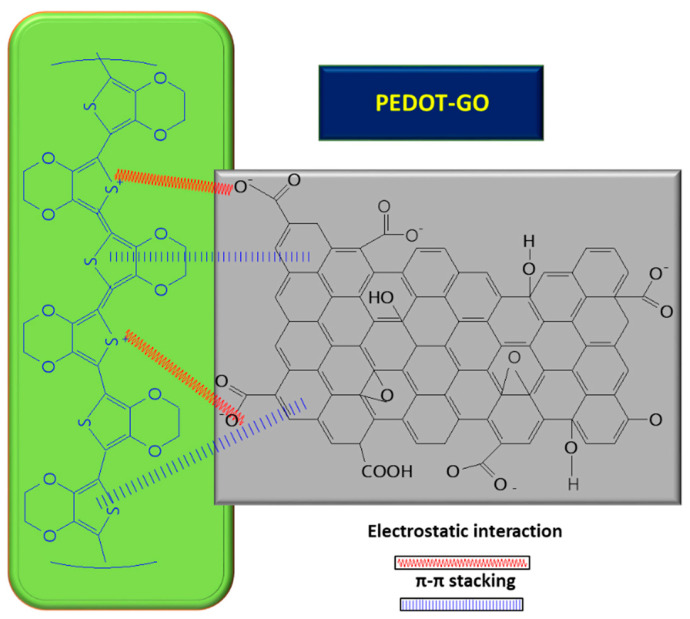
Interaction between GO and PEDOT, inside the composite used for UA detection.

**Figure 5 molecules-28-00135-f005:**
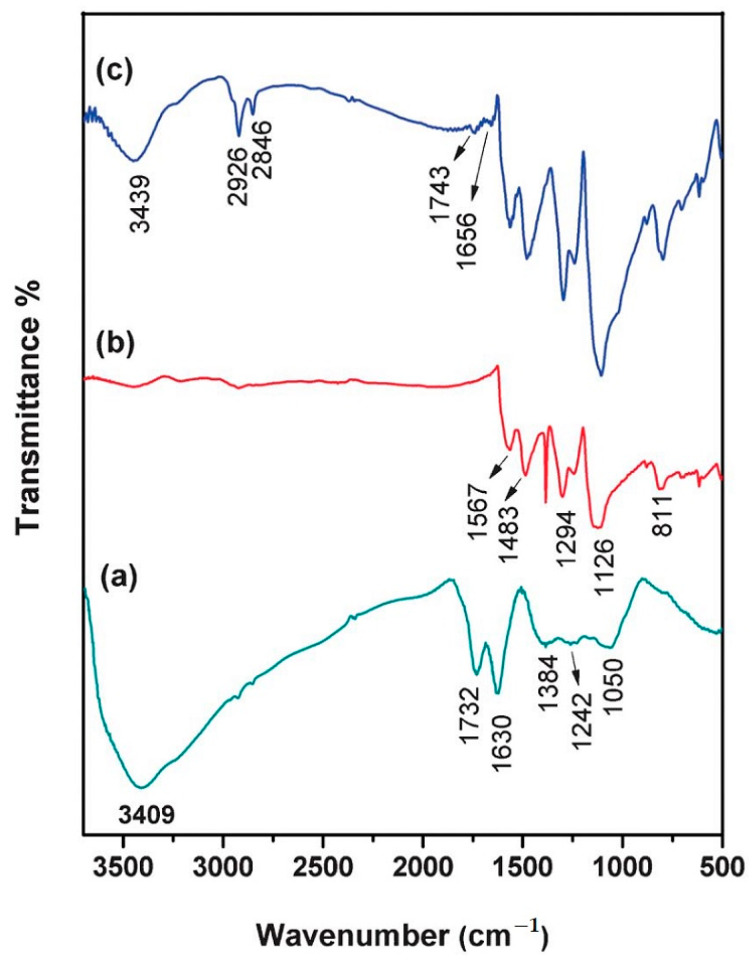
FTIR spectra of the compounds: (**a**) GO (green line); (**b**) PANI (red line); and (**c**) PANI-GO (blue line). Reproduced with the permission from ref. [63]; Copyright 2022 Springer.

**Figure 6 molecules-28-00135-f006:**
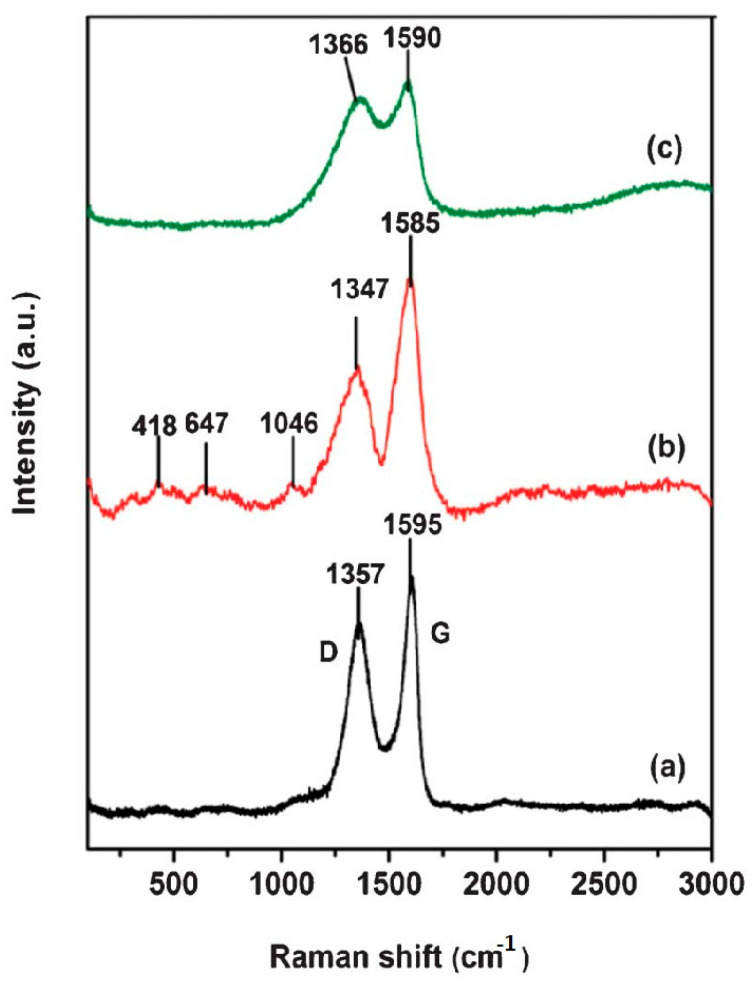
Raman spectra of the compounds: (**a**) GO (black line); (**b**) PANI (red line); and (**c**) PANI-GO (green line). Reproduced with the permission from ref. [63]; Copyright 2022 Springer.

**Figure 7 molecules-28-00135-f007:**
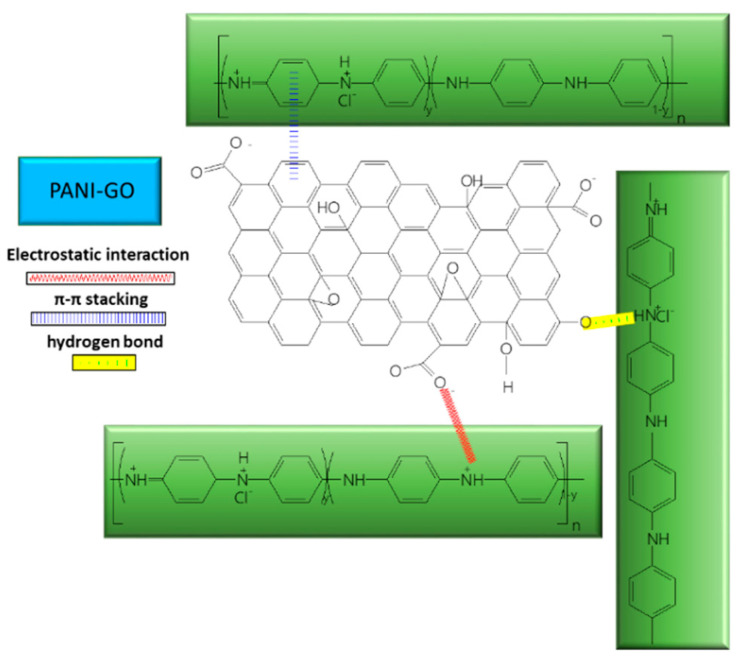
Interaction between GO and PANI in the composite used for UA detection.

**Figure 8 molecules-28-00135-f008:**
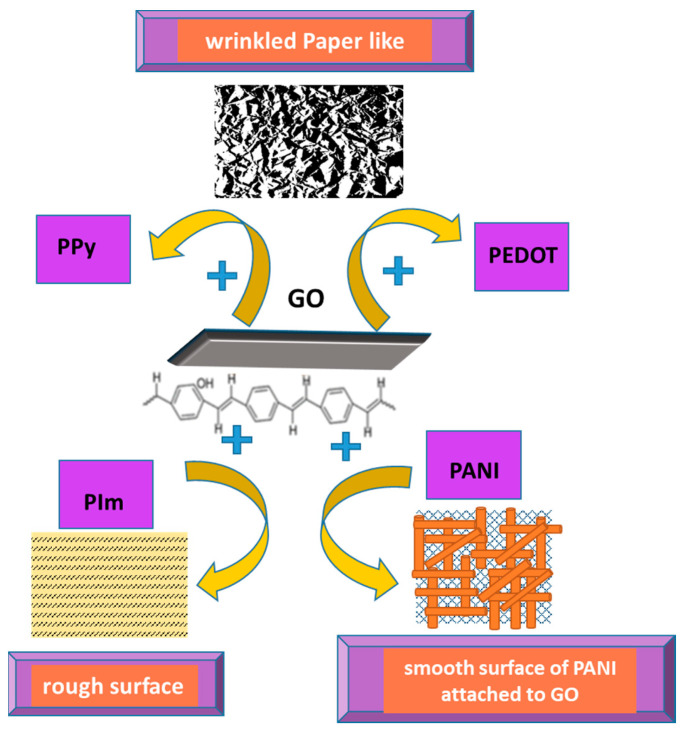
Different surface morphology of CP/GO composites, depending on the CP structure.

**Figure 9 molecules-28-00135-f009:**
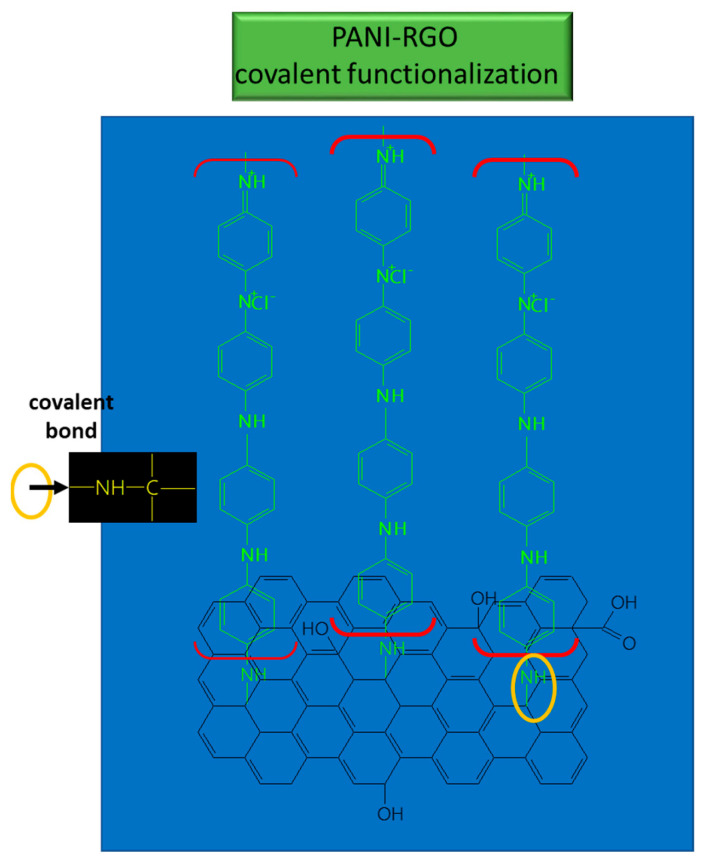
Interaction between RGO and PANI in the composite used for UA detection.

**Figure 10 molecules-28-00135-f010:**
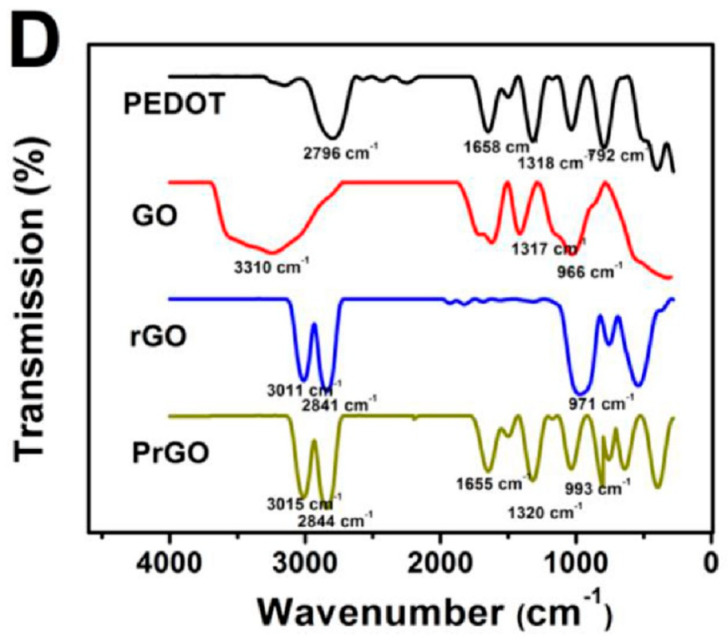
FTIR spectra of the compounds: PEDOT (black curve), GO (red curve), rGO (blue curve); and PrGO (dark yellow). Reproduced with permission from ref. [16]; Copyright 2022 MDPI.

**Figure 11 molecules-28-00135-f011:**
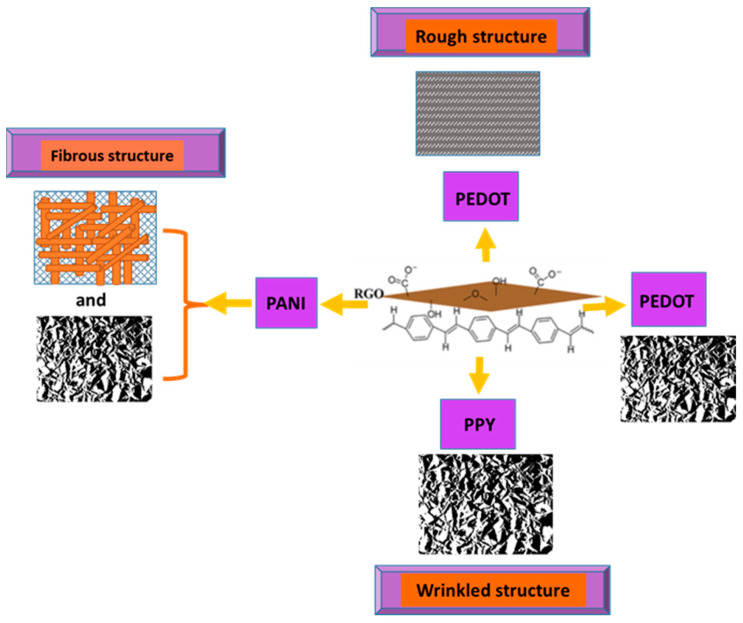
CP/RGO composites with different surface morphology, depending on the CP structure.

**Figure 12 molecules-28-00135-f012:**
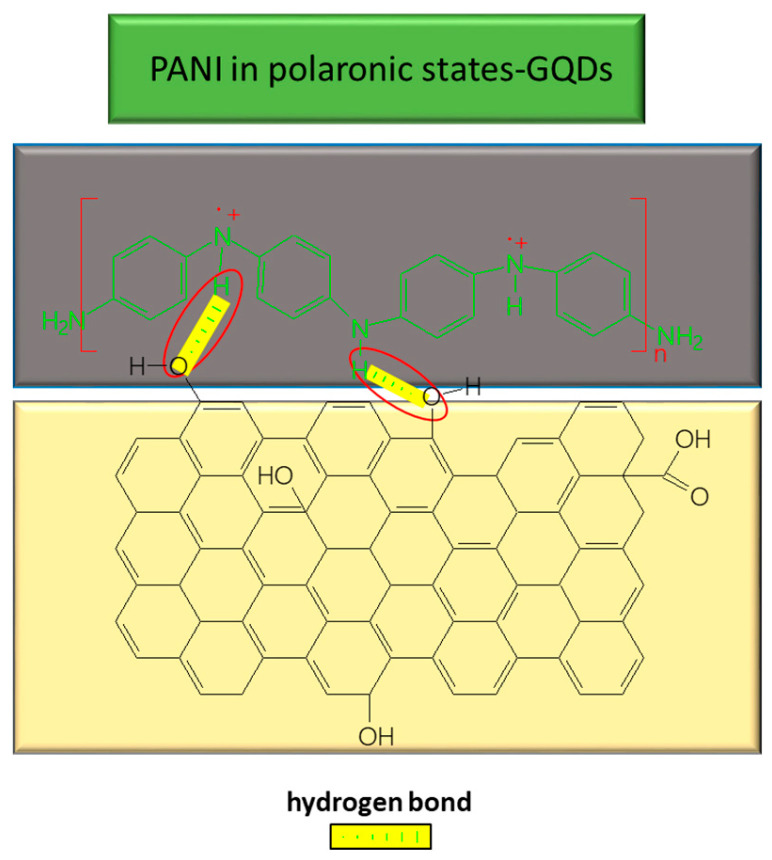
Interaction between GQDs and PANI in composite used for UA detection.

**Table 1 molecules-28-00135-t001:** Surface area and charge transfer resistance (R_ct_) of various electrodes.

Electrode	Surface Area (cm^2^)	Rct (Ω)
PEDOT-GO/GCE	0.1385	0.2
PEDOT/GCE	0.0787	77
GO/PEDOT/GCE	0.1027	300

**Table 2 molecules-28-00135-t002:** Synthesis methods and composites used for the preparation of electrodes for use in sensorial platforms, as well as their LOD.

Sensor (Composite/Electrode)	Method of Synthesis	Linear Range (µM)		LOD (µM)		Ref.
AuNPs/PPy–GO/CFP	Co-deposition through electro-polymerization	2–437	2–360-SD	1.39	1.68-SD	[6]
p-TPP/PPy–GO/GCE	Oxidative polymerization	5–200	-	1.15	-	[5]
PANI–GO	In situ chemical polymerization	2–18	-	0.2	-	[66]
PEDOT–GO/ITO PADS	Co-deposition	2–1000	-	0.75	-	[8]
PImox–GO/GCE	Polymerization		3.6–249.6-SD	-	0.59-SD	[9]
PEDOT/GO/GCE	Co-electrochemical deposition	40–240-SD		-	10-SD	[59]

SD: simultaneous detection: detection of UA in the presence of other biomolecules.

**Table 3 molecules-28-00135-t003:** Composite materials used as sensors for the detection of UA and their LODs.

Composite Material	Method of Synthesis	LR of Concentration (µM)		LOD (µM)		References
PPy/RGO (rGO/Pd@PPy NPs)	Co-polymerization	1.4–219	-	0.047	-	[29]
OPPy/RGO	Electrodeposition followed by over-oxidation of electropolymerized PPy	Used mainly for DA detection, the parameters for UA signal were also investigated	-	LR and LOD for UA were not reported in this paper, but the oxidation peak of UA was recorded at: OPPy/ERGO/GCE 15 µA, ERGO/GCE 3.1 µA, OPPY/GCE 1.1 µA, and GCE 0.8 µA	-	[71]
PANI/RGO (ZnO/PANI/RGO/GCE)	Electrochemical deposition	0.1–1000 100–1000	0.5–90 µM-SD	0.042	0.12 µM-SD	[4]
PANI/Fe_2_O_3_-SnO_2_/rGO (PFSG) ternary nanocomposite	Two-step hydrothermal	5–300-SD	-	1.6-SD	-	[27]
PEDOT/RGO	Electrodeposition	1–300	-	0.19 (S/N = 3)	-	[16]
MnO_2_/PEDOT/RGO (PrGO/MnO_2_)	Co-deposition through electropolymerization	0.3–80	-	0.05	-	[72]

**Table 4 molecules-28-00135-t004:** LOD of sensors based on GQDs used for the UA detection.

Electrode	Analytes	LOD	Reference
Au-PtNPs/GQD/GCE	AA, DA, UA, RT	1 × 10^−9^ M	[30]
GQDs/IL-SPCE	UA	3 × 10^−8^ M	[45]
GQDs/IL-SPCE	AA, DA, UA	2 × 10^−8^ M	[45]
SnO_2_/PANI/N-GQD	DA mixed with AA and UA	2.2 × 10^−7^ M	[116]
GQDs-doped PEDOT/GCE	AA, DA, UA	4.1 × 10^−6^ M	[117]

## Data Availability

Not applicable.

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
