# Peer review of "Functionalization of Graphene Derivatives with Conducting Polymers and Their Applications in Uric Acid Detection"

_molecules, 2022, doi:10.3390/molecules28010135_

Round 1
Reviewer 1 Report (Previous Reviewer 2)
The authors made changes to the manuscript as suggested by the reviewers.
Author Response
Thank you for your comment! The manuscript was checked for English language through the MDPI service. The changes are highlighted with track changes.
Reviewer 2 Report (New Reviewer)
The authors in the manuscript review the functionalization of graphene with conducting polymers and their applications in the uric acid detection. However, the organization, style, and writing of the manuscript are difficult to understand. I cannot recommend the work to be published in Molecules.
Author Response
The manuscript has been revised according to the suggestions and observations provided by the reviewer and checked for English language through the MDPI service. The changes are highlighted with track changes.
Reviewer 3 Report (New Reviewer)
In this work, Mirela Văduva et al. study about recent progress of conducting polymer/graphene based material and application in uric acid detection. After considering manuscript, we proposed that manuscript should be modified before submission. The following comments should be concerned:
1. The classification of material in Introduction section should be updated to clarify the motivation. If the author want to separate each conducting polymer, a comparison of their advantage/disadvantage have to mention. And if such comparison is difficult, merging all of CP introduction should be considered.
2. In second section, the author set name as "synthesis and optical characterization...". Indeed, when "optical characterization" is wrote, reader would assign it in typical characterizations such as UV/Vis spectra, PhotoLuminescence, etc. Meanwhile, in the manuscript, the author described about FT-IR, XPS, Raman which are chemical and vibration properties of materials. For example, in 2.1 and 2.2 section, no optical characterization was mentioned. In addition, which characterization is the most importance? In 2.1 and 2.2, reader would have a feeling that FT-IR is the key characterization for composite, but in 2.3, the FT-IR was neglected and UV-Vis was a sole characterion.
3. The name of 2.4 section can make reader misunderstanding.
4. Is it reasonable that there is only 4.1 subsection in 4. section (no 4.2 subsection)?
5. All of figures should be checked again to correlate the typos. Also, the mention about re-printed with permission should be revised.
Author Response
The classification of material in Introduction section should be updated to clarify the motivation. If the author want to separate each conducting polymer, a comparison of their advantage/disadvantage have to mention. And if such comparison is difficult, merging all of CP introduction should be considered.
Author’s reply: Thank you for your detailed response, we greatly appreciate all the suggestions and recommendations, they were very useful.
Considering all the recommendation given by the reviewer, the following changes have been made: at the introduction several information meant to complete the comparison of the advantages /disadvantages of each conducting polymer used in composite synthesis have been added; there were kept only the information regarding the main conductive polymers: PANI, PEDOT, PIm and PPy , all the other mentions of polyaminoacids and poli (β-cyclodextrine) have been removed from the entire manuscript.
In second section, the author set name as "synthesis and optical characterization...". Indeed, when "optical characterization" is wrote, reader would assign it in typical characterizations such as UV/Vis spectra, PhotoLuminescence, etc. Meanwhile, in the manuscript, the author described about FT-IR, XPS, Raman which are chemical and vibration properties of materials. For example, in 2.1 and 2.2 section, no optical characterization was mentioned. In addition, which characterization is the most importance? In 2.1 and 2.2, reader would have a feeling that FT-IR is the key characterization for composite, but in 2.3, the FT-IR was neglected and UV-Vis was a sole characterion.
Author’s reply: Thank you for your detailed response, we greatly appreciate the suggestion and recommendation, they were very helpful. Indeed, optical characterization is represented by investigations performed especially through UV-Vis absorption spectroscopy, IR and Raman spectroscopy, Photoluminescence but from all the reported papers where composite based on conductive polymers and carbon structures have been tested in uric acid detection, there are few mentions to optical characterization tools, the electrochemical characterization being the part that prevails. But, considering your observation we choose to change the name of the subsections from "synthesis and optical characterization..." to “Chemical and vibrational properties of ......”, the changes have been done in the following sections: 2.1, 2.2 and 2.3.
The name of 2.4 section can make reader misunderstanding.
Author’s reply: The subsection 2.4 was entirely removed (the section which contained the discussion about poly-aminoacids and graphene derivatives) and the remained subsection were renumbered accordingly. Figures 8 and 11 have been modified accordingly.
Is it reasonable that there is only 4.1 subsection in 4. section (no 4.2 subsection)?
Author’s reply: Thank you for the observation and suggestion. At the section 4, the subsection was removed so the entire text remained under the same title reflecting all the aspects from synthesis to vibrational properties and sensor performance in uric acid detection. There have been also inserted additional information regarding the GQDs-CPs vibrational properties, the changes have been done using track changes and they are highlighted using red font.
All of figures should be checked again to correlate the typos. Also, the mention about re-printed with permission should be revised.
Author’s reply: The word “reprinted” used inappropriately was replaced with “reproduced”.
The manuscript was checked for English language through the MDPI service. The changes are highlighted with track changes.
Round 2
Reviewer 2 Report (New Reviewer)
The manuscript has been thoroughly polished by the authors to improve its quality. I am now satisfied to recmmend it be published in Molecules.
Reviewer 3 Report (New Reviewer)
This work is good enough for publication.
This manuscript is a resubmission of an earlier submission. The following is a list of the peer review reports and author responses from that submission.
Round 1
Reviewer 1 Report
This mauscript can be published in this journal.
Author Response
We are very grateful for the time and effort that you invested in order to read and analysis our work.
Reviewer 2 Report
The review is well written with good scientific foundations on the proposed topic. The only criticism is the number of figures. The authors could add more explanatory figures on the formation of the graphene/polymer composite. As well as some examples of electrochemical response to uric acid.
Author Response
The only criticism is the number of figures. The authors could add more explanatory figures on the formation of the graphene/polymer composite. As well as some examples of electrochemical response to uric acid.
First of all we are very grateful for the time and effort that you invested in order to read and analysis our work. We greatly appreciate the suggestion regarding to add more explanatory figures to the main text. The added figures depict the functionalization process of the main conductive polymers, PEDOT, PANI and PPy and the oxidation reaction of uric acid which take place on the sensor surface. The figures were embedded into the main text, for each chapter discussed there. Additional changes have been made in the manuscript using track changes.
Reviewer 3 Report
Ideas between sections are not connected, and ideas discussed within the section are also not compared, contrasted, or summarized. Presents individual gist of many papers.
No figures, only two schemes in the whole paper. It would be more interesting to include figures to organize the information.
Table 3 caption repeated twice. page 19 and 23
Line 1027, page 23 was not continuation from previous paragraph and is not a start of new paragraph. Confuse readers.
Conclusions are not giving an overall summary and future challenges, also discussing noble metal nanocomposites in conclusion, seems like deviating from the scope of paper that is conducting polymer and graphene nanocomposites.
Author Response
We are very grateful for the time and effort that you invested in order to read and analysis our work. We greatly appreciate all the observations, recommendations and critical analysis of the paper. All the efforts brought together will finally lead to a more understandable and complete manuscript whose objective is to enrich the scientific community with a valuable content.
In the following, the responses to observations and suggestion provided by the reviewer are presented:
- Ideas between sections are not connected, and ideas discussed within the section are also not compared, contrasted, or summarized.
R. Additional sentences in order to link better the chapters were added into the main text. Moreover, the conclusions were completed by adding a comparative analysis between the performance of the electrodes modified with composites considering the best LOD and respectively the best LR reported, both in individual and simultaneous detection of UA, the latest being the most important in evaluating the sensors by direct confrontation with similar composition as within the real biologic samples collected from patients. The corresponding text was embedded also into the main manuscript.
- Presents individual gist of many papers. No figures, only two schemes in the whole paper. It would be more interesting to include figures to organize the information
R. This aspect could be due to the large number of characterization results which were mentioned into the text in order to provide a more understandable explanation of the functionalization process between the CPs and graphene structures. The functionalization type influences the electrochemical performance of the sensor, but it is not necessarily detailed or even mentioned in many articles. Because in papers dedicated to electrochemical sensors used in UA detection, the focus is mainly on the electrochemical parameters of the sensor, the part dedicated to the composite characterization is often neglected. The purpose of this paper is therefore to bring to light the aspects of optical characterization of the interaction between the composite components in order to give the readers a clearer way to understand the mechanism of the sensors in interaction with the analytes. More explanatory figures have been added as suggested, describing the functionalization process of the main conductive polymers, PEDOT, PANI and PPy with graphene derivatives. The figures were embedded into the main text, for each chapter discussed there.
- Table 3 caption repeated twice. page 19 and 23. R.The changes have been made as suggested.
- Line 1027, page 23 was not continuation from previous paragraph and is not a start of new paragraph. Confuse readers. R.The changes have been made as suggested.
- Conclusions are not giving an overall summary and future challenges
R. The conclusions were completed by adding a comparatory analysis between the performance of the electrodes modified with composites considering the best LOD and respectively the best LR reported, both in individual and simultaneous detection of UA, the latest being the most important in evaluating the sensors by direct confrontation with similar composition as within the real biologic samples collected from patients. The corresponding text was embedded also into the main manuscript.
- also discussing noble metal nanocomposites in conclusion, seems like deviating from the scope of paper that is conducting polymer and graphene nanocomposites.
R. As suggested the sentences which refers to the ternary composites have been removed from the conclusions. Furthermore, additional information highlighting the benefits and drawbacks of modifying the electrode with this kind of composites have been revealed. All the changes have been made in the manuscript using track changes.
Reviewer 4 Report
This manuscript entitled "Functionalization of graphene derivatives with conducting polymers and their applications in the uric acid detection" by et Văduva al. highlights RGO and interaction with uric acid. The authors are expected to provide more focus on the title as there are plethora of reviews on RGO and CPs reported extensively. The review is expected to be enriching to the audience in terms of additional advancement. I felt a lack in that. More focus have been given on the synthesis and optoelectronic properties of the RGO, CPs, composites etc. Thus this current manuscript should be modified through a major revision. I suggest the author to focus more on the carbon materials and Uric acid interaction and highlight on critical analyses of this phenomena. Subsequently, the outlook has to be modified as well.
Author Response
First of all we are very grateful for the time and effort that you invested in order to read and analysis our work. We greatly appreciate all the observations, recommendations and critical analysis of the paper. All the efforts brought together will finally lead to a more understandable and complete manuscript whose objective is to enrich the scientific community with a valuable content.
In the following the responses to observations and suggestion provided by the reviewer are presented:
- This manuscript entitled "Functionalization of graphene derivatives with conducting polymers and their applications in the uric acid detection" by et Văduva al. highlights RGO and interaction with uric acid. The authors are expected to provide more focus on the title as there are plethora of reviews on RGO and CPs reported extensively.
R. More explanatory figures have been added as suggested, describing the functionalization process of the main conductive polymers, PEDOT, PANI and PPy with graphene derivatives. The figures were embedded into the main text, for each chapter discussed there.
- The review is expected to be enriching to the audience in terms of additional advancement. I felt a lack in that. More focus has been given on the synthesis and optoelectronic properties of the RGO, CPs, composites etc.
R. According to the title the manuscripts deals with the topic of composites based on graphene derivatives and conducting polymer used for uric acid electrochemical detection. The graphene derivatives range was covered by discussing about GO, RGO and graphene quantum dots functionalized with different conducting polymers. Functionalization process could only be discussed in terms of optical characterization (including here technique as UV-Vis, Raman and FTIR spectroscopy followed by surface analyzing technique such as XRD, XPS, SEM…etc) related to synthesis method. Considering those informations mentioned above together with results obtained through electrochemical characterization methods such as CV, EIS and others, we consider that the manuscript addresses solid and complex the subject proposed within the title.
- Thus this current manuscript should be modified through a major revision. I suggest the author to focus more on the carbon materials and Uric acid interaction and highlight on critical analyses of this phenomena.
R. In order to better summarize all the results discussed within the manuscript at each section dedicated to composites based on CPs/GO, CPs/RGO and respectively CPs/QDTs, used in UA detection, many data, compared in terms of sensor performance translated trough LOD and LR parameters, have been organized in 4 tables (see Tables 1-4 embedded into the main text). In the latest the LOD and LR were correlated with synthesis method and functionalization type between the CPs and the graphene derivatives and the resulted conclusion of their comparative analysis were added into the main text, at conclusions section.
- Subsequently, the outlook has to be modified as well.
R. The outlook has been modified by highlighting the drawbacks of the composites currently used in the development of sensors for electrochemical detection of biomolecules. Additional information was added suggesting the area where further improvements can be done. All the changes have been made in the manuscript using track changes.
Round 2
Reviewer 3 Report
The authors improved the manuscript, including 7 Figures, and 3 Tables, and extended the discussions. Also modified the summary and outlook section.
The figures were not informative. Example: Figure 1 and Figure 2, Figure 3 shows the interactions between GO, polymer, and analyte schematic. But no figures of characterization to show the evidence/proof for the interactions.
The text is a summary of many publications, but the ideas are not organized and presented clearly.
Example: Introduction: Line 74 describes methods of UA determination after that line 80 discusses the UA detection is important.
Example: page 26, section 2.3.3 Overview concerning the sensorial platforms based on GQDs, not clear what concerns were discussed under this section.
The author discusses the performance of specific material in the summary and outlook. The writing style is not meeting the review paper requirements.
Reviewer 4 Report
The authors have made suitable additions in the revised manuscript. I recommend publication at its current form.